# *Strongyloides* Hyperinfection Associated with *Enterococcus faecalis* Bacteremia, Meningitis, Ventriculitis and Gas-Forming Spondylodiscitis: A Case Report

**DOI:** 10.3390/tropicalmed5010044

**Published:** 2020-03-12

**Authors:** Liang En Wee, Su Wai Khin Hnin, Zheyu Xu, Lawrence Soon-U Lee

**Affiliations:** 1Singhealth Infectious Diseases Residency, Singapore 168753, Singapore; 2Department of Neurology, National Neuroscience Institute, Singapore 308433, Singapore; hninsuwai.khin@mohh.com.sg (S.W.K.H.); xu.zhe.yu@singhealth.com.sg (Z.X.); 3Department of Infectious Diseases, Tan Tock Seng Hospital, Singapore 308433, Singapore; Lawrence_lee@ttsh.com.sg or; 4Department of Medicine, National University of Singapore, Singapore 119077, Singapore

**Keywords:** *Strongyloides*, hyperinfection, enterococcus, emphysematous spondylodiscitis

## Abstract

An elderly Singaporean male with no travel history was hospitalized for fever and altered mental status. Blood cultures grew *Enterococcus faecalis*, and given a preceding history of steroid use and peripheral eosinophilia, *Strongyloides* hyperinfection was suspected. Stool specimens were positive for *Strongyloides stercoralis* larvae over four days, and larvae were also isolated in an early morning nasogastric aspirate specimen prior to initiation of ivermectin. A cerebrospinal fluid examination was consistent with partially treated bacterial meningitis and ventriculitis was demonstrated on neuroimaging. In view of a persistent fever, a further imaging evaluation was performed, which demonstrated bilateral pneumonia as well as the unusual finding of gas-forming emphysematous spondylodiscitis and left psoas abscesses. Despite the early suspicion of *Strongyloides* hyperinfection, commencement of appropriate antibiotics and anti-helminthics, microbiological clearance of bacteremia as well as clearance of *S. stercoralis* from the stool, the patient still succumbed to infection and passed away 11 days after admission.

## 1. Introduction

*Strongyloides* is common in tropical and subtropical regions, although the distribution of this intestinal nematode can occur worldwide. Chronically infected patients are at risk of developing *Strongyloides* hyperinfection, especially with immunosuppression and steroid use [1]. The unique autoinfective lifecycle of this intestinal nematode allows for infection to persist in the host for a prolonged duration [1,2]; thus, the index of suspicion for this condition needs to be high, particularly as epidemiological exposures may be distant in time. Early suspicion and detection of *Strongyloides* hyperinfection is crucial, given its high mortality rates up to 90% [1]. *Strongyloides* hyperinfection is characterized by the dissemination of gut flora into the bloodstream, typically leading to bacteremia, pneumonia and involvement of the central nervous system; concomitant bone and deep soft tissue infection is less commonly reported [1,2,3]. Here we present a case of *Strongyloides* hyperinfection associated with *Enterococcus faecalis* bacteremia, gas-forming spondylodiscitis, and psoas abscess—a less common presentation—in a patient with previous steroid use.

## 2. Case Report

An 82-year-old Singaporean Chinese male was admitted to our institution with one day’s history of generalized lethargy, malaise and fever. His pre-existing medical conditions included hypertension, hyperlipidemia and atrial fibrillation. On initial presentation, the patient was noted to have a fever of 38.2 °C, with stable vital signs. In the emergency department, he developed a generalized tonic–clonic seizure, which was aborted with intravenous lorazepam. On physical examination, the patient had an initial Glasgow Coma Scale (GCS) of E1V1M2. Gaze was central; the neck was supple. There was no rash. The white blood cell (WBC) count on presentation was elevated at 19.2 × 10^9^/L (normal 4.0–9.6) with 93.8% neutrophils, 2.2% lymphocytes and 1.3% eosinophils. The patient was empirically started on intravenous (IV) ceftriaxone, ampicillin and acyclovir to cover for meningoencephalitis, as well as phenytoin. Blood cultures from admission grew *E. faecalis*, which was susceptible to penicillin. 

On the third day of hospitalization, the patient was referred to infectious diseases for a further opinion as he still remained febrile despite the blood cultures from day two of hospitalization, demonstrating clearance of *E. faecalis* bacteremia. The repeat WBC count was still elevated at 20.4 × 10^9^/L (normal 4.0–9.6); however, significant peripheral eosinophilia was also noted, with 8.1% eosinophils, 83.8% neutrophils and 4.1% lymphocytes. The eosinophil count was 1.65 × 10^9^/L (normal 0.0–0.6). In light of the significant peripheral eosinophilia, suspicion of meningoencephalitis and isolation of a microorganism less typically associated with meningitis, stool specimens were sent for ova, cyst and parasite (OCP) examination, due to suspicion of hyperinfection with *Strongyloides stercoralis*.

On the fourth day of hospitalization, the lumbar puncture was successfully performed under radiological guidance. Cerebrospinal fluid (CSF) findings were in keeping with a partially treated pyogenic meningitis, with an elevated CSF protein of >2.00 g/L (normal 0.10–0.40), low CSF glucose of 2.3 mmol/L(normal 2.5–5.5) and a white cell count of 173 cells/microliter (normal 0–5); however, CSF cultures were negative, likely because of prior antibiotic treatment. Cytology showed a mixed inflammatory yield of mononuclear cells and neutrophils; polymerase chain reaction (PCR) tests for cytomegalovirus, herpes simplex virus, varicella zoster virus, human herpes-virus 6, enterovirus and parechovirus were all negative, and PCR tests for typical bacterial pathogens causing meningitis, including *E. coli, H. influenzae, L. monocytogenes, N. meningitidis, S. agalactiae, S. pneumoniae*, were also negative. Fungal cultures and acid-fast-bacilli smears and cultures were also negative. Intravenous acyclovir was stopped; IV ceftriaxone and ampicillin were continued. Subsequently, *Strongyloides stercoralis* larvae were detected on microscopic examination of a stool specimen on the fourth day of hospitalization, as well as on microscopic examination of early morning nasogastric aspirate specimens. Direct smear and formalin–ether concentration examinations were conducted on the stool samples. Multiple smears were made for each technique, followed by systematic microscopic examination.

Given the suspicion of hyperinfection with *Strongyloides stercoralis*, oral ivermectin was started at 200 mcg/kg/day daily and administered via a nasogastric tube. Additional history was obtained: The patient was born in Singapore and had grown up in a rural village in the 1930s with no other travel history. Other family members did not recollect any history of passing worms per-rectum in childhood. He previously worked in a lumber mill but had retired more than 30 years ago. In the 1970s, the patient had moved out of his village to live in an apartment block, and at the point of admission, was staying in an apartment block in western Singapore. He did not have any exposure to soil. Although prednisolone was stopped two months prior to his current admission, he had instead continued taking prednisolone up to the day of admission. For the past two decades, he was noted to have intermittent peripheral eosinophilia but did not undergo further evaluation (Table 1). On review of previous blood tests, he was first noted to have peripheral eosinophilia 17 years prior to presentation (eosinophil count of 0.81 × 10^9^/L); three years ago, when he was first diagnosed with adrenal insufficiency secondary to exogenous steroid usage, the peripheral eosinophil count still remained raised (eosinophil count of 3.88 × 10^9^/L). He continued to have intermittent eosinophilia on follow up. He was initiated on steroids for the adrenal insufficiency, while the endocrinologist had stopped prescribing prednisolone three months prior to presentation; it became apparent during medication reconciliation that the patient had inadvertently continued on both prednisolone 5 mg as well as hydrocortisone 10 mg once-daily up to the point of his admission.

On the fifth day of hospitalization, magnetic resonance imaging (MRI) of the brain demonstrated ventriculitis. There was no evidence of brain abscesses. On the seventh day of hospitalization, in view of persistent pyrexia, a computed tomography (CT) scan of the thorax, abdomen and pelvis was performed, which showed gas locules within the L3/L4 intervertebral disc with asymmetric thickening of the left psoas muscle, suggestive of a psoas abscess and concurrent spondylodiscitis (Figure 1a,b). There were also bilateral patchy airspace changes and moderate pleural effusions suggestive of pneumonia (Figure 1c) and small pericardial effusion. The transthoracic echocardiogram was negative for vegetations. In view of the prolonged hospitalization and persistent fever, his antibiotic coverage was changed to IV cefepime, metronidazole and ampicillin, to provide broader empiric nosocomial cover. Repeated blood cultures remained negative. An MRI scan of the lumbar spine showed left psoas myositis with phlegmon and small gas-containing abscesses. However, as the abscesses were small and the patient was coagulopathic, drainage of the abscesses was deemed infeasible. 

*Strongyloides stercoralis* was persistently detected on microscopic examination of daily stool specimens from the fourth to eighth day of admission. However, although *Strongyloides* was also detected on early morning nasogastric aspirates on the fourth day admission, after ivermectin was initiated, subsequent sampling of nasogastric aspirates was persistently negative. Microscopy of urine specimens for *Strongyloides* was also negative. *Strongyloides* IgG returned as positive; testing for antibodies against human immunodeficiency virus (HIV) and human T-lymphotrophic virus (HLTV) were negative. IgE levels were significantly elevated (1064 kU/L; normal < 113 kU/L). The patient was treated with daily ivermectin, and on Day 10 of admission was also given a single dose of 400 mg albendazole. However, he continued to deteriorate and passed away on Day 11 of admission. Blood cultures done just prior to his demise were negative, and two consecutive microscopic examinations of his stool, as well as sputum/early morning nasogastric aspirates, were negative for *Strongyloides stercoralis*. A final diagnosis of *Strongyloides* hyperinfection, in the context of exogenous steroid usage, complicated by *E. faecalis* bacteremia, bacterial meningitis/ventriculitis as well as gas-forming spondylodiscitis and a psoas abscess was made. 

## 3. Discussion

While *Strongyloides* is common in the tropical and subtropical regions, few cases have been reported in Singapore, a highly-urbanized city state. However, cases of *Strongyloides* hyperinfection have been reported in immunocompromised elderly individuals, who were most likely exposed in childhood prior to the rapid urbanization of Singapore in the 1960s–1970s [4,5,6]. Cases of *Strongyloides* hyperinfection have been previously reported in British veterans of the Second World War who were exposed during their service in Southeast Asia, demonstrating the long latent period and potential chronicity of infection [7]. In our patient, given his lack of a travel history and lack of recent soil exposure, we speculate that his infection was likely acquired earlier in life and had persisted for decades, with steroid exposure being a known risk factor for progression to *Strongyloides* hyperinfection [1,2]. He had a documented history of peripheral eosinophilia for almost two decades prior to his presentation. While the majority of patients with chronic *S. stercoralis* infection may have peripheral eosinophilia, in the situation of hyperinfection, eosinophil counts may potentially be suppressed [2]. In this patient, while the initial full blood count did not show eosinophilia, subsequent development of eosinophilia, the clinical suspicion of meningitis and isolation of *E. faecalis* from the blood prompted a search for *Strongyloides.* While *Strongyloides* hyperinfection is typically associated with Gram-negative bacteremia and meningitis, there have been case reports of hyperinfection associated with *Enterococcus* spp. [8,9,10]. 

Although our patient presented with the triad of bacteremia, central nervous system involvement and pneumonia typical of *Strongyloides* hyperinfection [1,2,3], he also had unusual features of gas-forming spondylodiscitis and psoas abscesses. Seeding of the spine might potentially have occurred through hematogenous spread of the bacteria, or through dissemination of the *Strongyloides* larvae. However, while larva currens and cutaneous manifestations of *Strongyloides* hyperinfection are well-documented [5], dissemination to deep soft tissue and spinal involvement is less commonly reported [1,2,3]. Gas-forming spondylodiscitis is an uncommon radiological finding [11,12,13,14,15]. Common causes of emphysematous osteomyelitis include anaerobes, Gram-negative organisms (e.g., *K. pneumoniae*, *E. coli*) and *Staphylococcus aureus* [12,13,14,15]; though, *E. faecalis* has previously been reported as a cause of gas-forming spinal epidural abscess [11]. In our patient, the only positive microbiology was *E. faecalis* isolated from the bloodstream. Gas-forming osteomyelitis was associated with a higher mortality rate compared to vertebral osteomyelitis with non-gas-forming organisms [12]. Despite early suspicion of *Strongyloides* hyperinfection, commencement of appropriate antibiotics, anti-helminthics and microbiological clearance of both *E. faecalis* from the bloodstream and clearance of *S. stercoralis* from the stool (demonstrated by two consecutive negative microscopic examinations of both stool and early morning nasogastric aspirates), our patient still succumbed to his infection. This highlights the high mortality associated with *Strongyloides* hyperinfection [1,2] and the high index of suspicion required to establish early diagnosis and institute appropriate treatment. 

## Figures and Tables

**Figure 1 tropicalmed-05-00044-f001:**
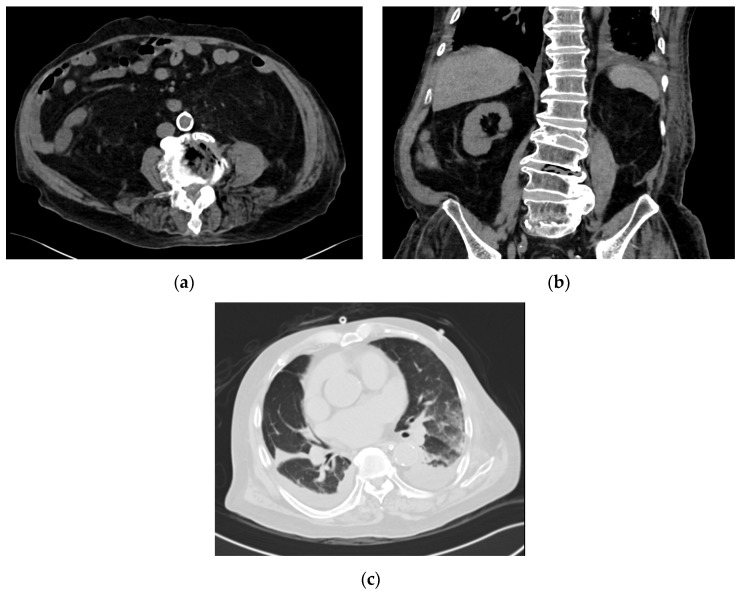
*Strongyloides* hyperinfection associated with gas-forming spondylodiscitis and *E. faecalis* bacteremia. (**a**) Transverse view of the computed tomography scan of the abdomen, demonstrating gas-forming spondylodiscitis. (**b**) Coronal view of the computed tomography scan of the abdomen, demonstrating gas-forming spondylodiscitis. (**c**) Bilateral pulmonary infiltrates noted on the computed tomography scan of the thorax.

**Table 1 tropicalmed-05-00044-t001:** Eosinophil trend, clinical events and history of steroid usage prior to presentation.

Date	2002	2004	March–April 2016				2018		January 2019	July 2019	December 2019				
Eosinophil count (×10^9^/L)	0.8	0.8	3.9	2.8	0.8	0.6	0.5	0.0	0.8	0.2	0.3	1.7	2.2	1.1	0.8
Eosinophil %	9.5	6.5	25.4	31.1	4.7	6.3	4.2	0.1	8.4	1.9	1.3	8.1	16.6	12	5.4
Event	Routine primary care visit	Admitted for bleeding gastric ulcer	Admitted for left upper limb cellulitis and deconditioning in March 2016; transferred to a community hospital for rehabilitation in April 2016. Had two documented episodes of E.coli urinary tract infection				Routine primary care visit	Routine primary care visit	Routine primary care visit	Routine primary care visit	Admitted in mid-December 2019 for fever and altered mental status, diagnosed with *Strongyloides* hyperinfection (stool positive from D4-D8 of admission) Started on daily oral ivermectin from D4-D11 of admission.				
Steroid usage	Not known	Traditional Chinese medicine	Started on hydrocortisone 20 mg once every morning, 10 mg once every evening for adrenal insufficiency on 16/3/16; dose subsequently reduced further to 10 mg daily				Remains on maintenance hydrocortisone 10 mg once daily			Switched to prednisolone 5 mg once daily from June 2019 and stopped from September 2019, but patient inadvertently took hydrocortisone 10 mg once daily and prednisolone 5 mg once daily up till December 2019 admission.					

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
