# Peer review of "Strongyloides Hyperinfection Associated with Enterococcus faecalis Bacteremia, Meningitis, Ventriculitis and Gas-Forming Spondylodiscitis: A Case Report"

_tropicalmed, 2020, doi:10.3390/tropicalmed5010044_

Round 1

Reviewer 1 Report

Abstract

Line 15: E. faecalis in full at first mention in the abstract.

Line 24: add a space between the . and stercoralis (S. stercoralis). Ditto elsewhere throughout the document (i.e. line 56 E.faecalis)

Line 39: E. faecalis in full at first mention in the Introduction/main body of the manuscript

Line 42: An

Line 49: °C

Line 65: Do you know what specific test/s they (presumably hospital or commercial lab) did for OCP? OCP covers a great many diagnostics, some of which are better for Strongyloides identification than others. Not overly crucial for a case study, but perhaps a more sensitive/specific to strongyloides test would have picked up infection in the final two stool specimens. Plus researchers often like to know what tests are being used!

Discussion (more a comment for a copy editor) – text needs to be justified as per rest of document

Author Response

We thank the reviewer for the kind comments. Please find below our point-by-point reply to the comments:

Line 15: E. faecalis in full at first mention in the abstract.

This has been done.

Line 24: add a space between the . and stercoralis (S. stercoralis). Ditto elsewhere throughout the document (i.e. line 56 E.faecalis)

This has been done.

Line 39: E. faecalis in full at first mention in the Introduction/main body of the manuscript

This has been done

Line 42: An
This has been done

Line 49: °C
This has been done

Line 65: Do you know what specific test/s they (presumably hospital or commercial lab) did for OCP? OCP covers a great many diagnostics, some of which are better for Strongyloides identification than others. Not overly crucial for a case study, but perhaps a more sensitive/specific to strongyloides test would have picked up infection in the final two stool specimens. Plus researchers often like to know what tests are being used!

We thank the reviewer for the feedback. We have included the following statements to clarify:
"Direct smear and formalin-ether concentration examinations were conducted on the stool samples. Multiple smears were made for each technique, followed
by systematic microscopic examination."

Reviewer 2 Report

This is a well written and well presented manuscript. Case description, diagnosis and discussion are well articulated. There are few minor concerns, which are mentioned below:

  1. In Introduction "particularly as epidemiological exposures may be distant in space and time." - What does this line mean
  2. in Case Report, Strongyloides description came at the end, which is the focus of the case report. Rearrangement of the section can easily address this issue.
  3. Figures are missing from the manuscript.

Author Response

We thank the reviewer for the kind feedback. Please find below our point-by-point response to the comments:

In Introduction "particularly as epidemiological exposures may be distant in space and time." - What does this line mean

We have corrected the phrase to:
"The unique autoinfective life cycle of this intestinal nematode allows for infection to persist in the host for a prolonged duration; (1-2) thus the index of suspicion for this condition needs to be high, particularly as epidemiological exposures may be distant in space and time. "
We thank the reviewer for the opportunity to clarify.

in Case Report, Strongyloides description came at the end, which is the focus of the case report. Rearrangement of the section can easily address this issue.

We thank the reviewer for the suggestion.
Though the events are presented in a chronological fashion, we have rearranged the case report such that the 1st paragraph is a brief summary of his presenting complaint, and the suspicion of Strongyloides is highlighted from the 2nd para onwards.

Figures are missing from the manuscript.
We apologise if the figures were inadvertently missed out- we append them with this manuscript.